# Widespread signatures of natural selection across human complex traits and functional genomic categories

Jian Zeng[1✉], Angli Xue[1], Longda Jiang[1], Luke R. Lloyd-Jones [1], Yang Wu [1], Huanwei Wang [1], Zhili Zheng[1], Loic Yengo [1], Kathryn E. Kemper[1], Michael E. Goddard[2,3], Naomi R. Wray [1,4], Peter M. Visscher [1] & Jian Yang [1,5,6✉]

Understanding how natural selection has shaped genetic architecture of complex traits is of importance in medical and evolutionary genetics. Bayesian methods have been developed using individual-level GWAS data to estimate multiple genetic architecture parameters including selection signature. Here, we present a method (SBayesS) that only requires GWAS summary statistics. We analyse data for 155 complex traits (n = 27k–547k) and project the estimates onto those obtained from evolutionary simulations. We estimate that, on average across traits, about 1% of human genome sequence are mutational targets with a mean selection coefficient of ~0.001. Common diseases, on average, show a smaller number of mutational targets and have been under stronger selection, compared to other traits. SBayesS analyses incorporating functional annotations reveal that selection signatures vary across genomic regions, among which coding regions have the strongest selection signature and are enriched for both the number of associated variants and the magnitude of effect sizes.

[1] Institute for Molecular Bioscience, The University of Queensland, Brisbane, QLD, Australia. [2] Faculty of Veterinary and Agricultural Science, University of Melbourne, Parkville, VIC, Australia. [3] Biosciences Research Division, Department of Economic Development, Jobs, Transport and Resources, Bundoora, VIC, Australia. [4] Queensland Brain Institute, The University of Queensland, Brisbane, QLD, Australia. [5] School of Life Sciences, Westlake University, Hangzhou, Zhejiang, China. [6] Westlake Laboratory of Life Sciences and Biomedicine, Hangzhou, Zhejiang, China. ✉email: j.zeng@uq.edu.au; jian.yang@westlake.edu.cn

The joint distribution of SNP effect size and minor allele frequency (MAF) is an essential component of the genetic architecture of human complex traits and is influenced by natural selection[1]. A negative relationship between effect size and MAF is a signature of negative (or purifying) selection[2,3], which prevents mutations with large deleterious effects becoming frequent in the population. Understanding how natural selection has shaped genetic variation helps researchers to improve experimental designs of genetic association studies[4] and the estimation of SNP-based heritability (the proportion of phenotypic variance explained by the SNPs)[5–9]. Inference on natural selection is also a critical step towards the understanding of the genetic architecture of complex traits. For instance, the theory of negative selection[10] explains why the effects of common variants identified by genome-wide association studies (GWAS) are unlikely to be large[11,12].

We have recently developed a Bayesian method (BayesS) to estimate the effect size–MAF relationship, which was considered as a free parameter ($S$) in the model[13]. We detected negative $\hat{S}$ for a number of complex traits in humans, highlighting an important role of negative selection in shaping the genetic architecture, consistent with the findings from other studies based on genome-wide variance estimation approaches[7,11,14,15]. The BayesS model also allows us to estimate the SNP-based heritability and polygenicity (the proportion of SNPs with nonzero effects) to better describe the genetic architecture for a trait. The application of BayesS has been restricted to GWAS samples with individual-level genotypes but for most common complex diseases, only summary-level data are publicly available. Moreover, despite the implementation of a parallel computing strategy[13], it remains computationally challenging to run BayesS for biobank-scale data, as the computing resource required increases linearly with the number of individuals or SNPs.

In this study, we enhance the BayesS model such that the analysis only requires GWAS summary statistics and a sparse linkage disequilibrium (LD) correlation matrix from a reference sample. This method (referred to as Summary-data-based BayesS or SBayesS) opens an opportunity to disentangle the genetic architecture of complex traits (including diseases) using publicly available data sets of the largest sample sizes to date, with merely a small fraction of the computational resource required for BayesS. We perform extensive analyses to benchmark between SBayesS and BayesS, and apply the SBayesS methods to GWAS summary statistics from the full release of the UK Biobank[16] (UKB) data and other published studies[17–25], followed by time-forward simulations[26] for evolutionary inference and SBayesS analyses that incorporated functional genomic annotation data. We detect widespread signatures of negative selection in the genetic architecture across 155 complex traits with a predicted mean selection coefficient of ~0.001 and a predicted mean proportion of human genome sequence being mutational targets of ~1%, among which common diseases show a relatively higher mean selection coefficient and a relatively smaller number of mutational targets. Meta-analysis across traits reveals differential signatures of negative selection across functional genomic regions, among which coding regions have the strongest selection signature and are enriched for both trait-associated variants and those with large effect sizes.

## Results

### Method overview
BayesS is a method that can estimate three key parameters to describe the genetic architecture of complex traits by a Bayesian mixed linear model[13], namely SNP-based heritability ($h^2_{SNP}$), polygenicity ($\pi$) and the relationship between MAF and effect size ($S$), all of which are defined with respect to a certain set of SNPs (see the definition of $h^2_{SNP}$ as an example[5]). BayesS requires individual-level data. SBayesS is an extension of BayesS, but only requires GWAS summary statistics of the SNPs and LD information from a reference sample (see the "Methods" section, Supplementary Note and Supplementary Fig. 1). We compute pairwise LD correlations between SNPs located on the same chromosome from a reference sample and remove correlations that can be attributed to sampling variation by a chi-squared test, resulting in a sparse LD matrix (see the "Methods" section). In addition, we model analytically the sampling variance of LD estimates as part of the residual variance and allow the estimate of residual variance to vary across SNPs (Supplementary Note). Compared to BayesS, SBayesS not only addresses the barrier of data sharing as it does not require individual-level data, but also substantially increases the computational efficiency because of the use of sparse LD matrix and a different updating strategy in the MCMC sampling (Supplementary Note). These features allow SBayesS to be scalable to data with millions of SNPs regardless of the discovery GWAS sample sizes. To examine the convergence of MCMC, we provide a GCTB-SBayesS implementation of the Gelman–Rubin statistic[27] which compares the variation between and within multiple chains with different starting values of the model parameters (see the "Methods" section). Convergence is only concluded if all the three key parameters converge, which may not occur if the LD matrix from a reference sample is too divergent from that of the GWAS sample, or if the summary statistics are generated from a GWAS with low power or contain uncorrected population stratification, poor imputation or other errors such as misreported per-SNP sample size and allele frequency.

In light of recent studies[11,28], which point out a possible lack of fit of a point–normal mixture model to some traits, we further extended SBayesS to a multi-component mixture model (referred to as SBayesRS), following the framework of SBayesR[29]. In SBayesRS, each SNP effect is assumed to have a mixture of a point mass at zero and three normal distributions with mean zero and variances that differ by a factor of 10 (see the "Methods" section). This flexible prior accounts for a more complex genetic architecture with a spectrum of very small to very large effect sizes. The $S$ parameter and overall polygenicity are estimated based on the SNPs across all nonnull mixture components.

To better understand the variability of regional genetic architecture in different parts of the genome, we incorporate functional genomic annotations into SBayesS to allow the three key parameters to vary in different annotation categories, e.g., coding, regulatory and repressed regions. We performed the functional partitioning SBayesS analysis (denoted SBayesS-strat) based on a two-component model that fitted SNPs in one annotation as the first component and the rest of the SNPs as the second component (see the "Methods" section). During MCMC sampling, the enrichment of a parameter in an annotation category is computed as the ratio of the sampled value of the parameter in the category to that for the whole genome (see the "Methods" section).

### Benchmarking SBayesS with BayesS
We ran both SBayesS and BayesS with ~1.1 million HapMap3 SNPs with MAF $\geq 0.01$ for 18 quantitative traits ($n > 100k$) as analysed in Zeng et al. [13]. We used the HapMap3 SNPs as they were optimised to tag common genetic variants[30] and are widely used in the literature which improves the comparability of our results with those generated using published GWAS summary statistics. Hence, the reported parameters are specific to this SNP set. For ease of computation, we used unrelated individuals of European ancestry from the interim release of the UKB data for the BayesS analysis (maximum $n = 120k$ across traits) and the same data to generate GWAS summary statistics for the SBayesS analysis. We show in

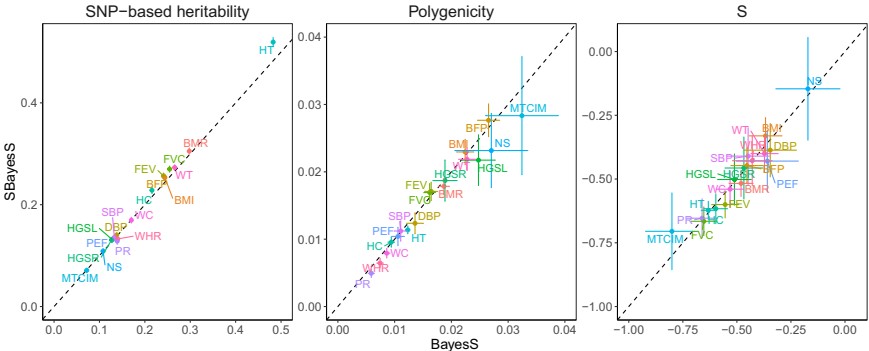

**Fig. 1 Benchmarking SBayesS with BayesS using the same data for 18 UKB traits.** Three genetic architecture parameters were compared, i.e., SNP-based heritability, polygenicity (defined as proportion of SNPs with nonzero effects) and S (defined as relationship between MAF and effect size), based on the unrelated individuals of European ancestry in the interim release of the UKB data (max $n = 120k$) and ~1.1 million HapMap3 common SNPs (MAF > 0.01). The sparse LD matrix used in SBayesS was computed from a random sample of 50k unrelated individuals from the full UKB cohort at a chi-squared threshold of 10 (corresponding to a LD $r^2$ threshold of $2 \times 10^{-4}$). Data are presented as posterior means ± posterior standard errors. The traits are indicated by different colours labelled with their acronyms. BMR basal metabolic rate, BMI body mass index, BFP body fat percentage, DBP diastolic blood pressure, FEV forced expiratory volume, FVC forced vital capacity, HGSL hand grip strength (left), HGSR hand grip strength (right), HCadjBMI hip circumference adjusted for BMI, HT height, MTCIM mean time to correctly identify matches, NS neuroticism score, PEF peak expiratory flow, PR pulse rate, SBP systolic blood pressure, WCadjBMI waist circumference adjusted for BMI, WHRadjBMI waist-hip ratio adjusted for BMI, WT weight.

Fig. 1 that the correlation between the SBayesS and BayesS estimates for all of the three genetic architecture parameters was close to one across traits (Pearson correlation $r = 0.998$ for $h^2_{\mathrm{SNP}}$, 0.985 for $\pi$ and 0.965 for $S$).

We performed additional sensitivity analyses to investigate the impact of the sparsity of LD matrix, the SNP panel, the choice of reference sample and the reference sample size on the performance of SBayesS. We found that SBayesS was robust to different chi-squared thresholds used for LD filtering (Supplementary Fig. 2) and gave consistent results with BayesS regardless of whether using HapMap3 (Fig. 1) or UKB Axiom array panel (Supplementary Fig. 3a). The analysis using HapMap3 SNPs tended to give slightly lower $\hat{h}^2_{\mathrm{SNP}}$ and $\hat{\pi}$ but stronger signals of S for both SBayesS and BayesS (Supplementary Fig. 3b, c), possibly due to the under-representation of low-frequency SNPs in HapMap3 panel in comparison with the UKB Axiom array panel (Supplementary Fig. 4). There was negligible difference in parameter estimates when the LD reference sample size ($n_{\mathrm{ref}}$) decreased from 50k to 20k but notable inflation in $\hat{h}^2_{\mathrm{SNP}}$ and $\hat{S}$ when $n_{\mathrm{ref}}$ further decreased to 4k (Supplementary Fig. 5), suggesting that the LD reference sample size cannot be too small relative to the GWAS sample size. Given a constant reference sample size ($n_{\mathrm{ref}} = 50k$), we ran GWAS with sample sizes $n_{\mathrm{gwas}} = 120k$ and 350k and found highly consistent parameter estimates between SBayesS and BayesS regardless of $n_{\mathrm{gwas}}$ (Supplementary Fig. 6). As expected, the $\pi$ estimate from either SBayesS or BayesS increased with larger $n_{\mathrm{gwas}}$ because of the increased power to detect small effects, consistent with the observation from an independent prior study[28]. Since $h^2_{\mathrm{SNP}}$ and S were estimated based on the SNPs with nonzero effects, the estimators of these parameters were also to some extent sample size dependent (see Supplementary Note for more discussion). Furthermore, with both $n_{\mathrm{ref}} = 50k$ and $n_{\mathrm{gwas}} = 300k$ held constant, we benchmarked BayesS and SBayesS in a few different scenarios where the LD reference was a subset of the GWAS sample, an independent sample from the same or slightly different population, or a sample of different ancestry. The performance of SBayesS was almost independent of the overlap between the GWAS and LD reference samples as long as they are from the same population but started to deteriorate when the genetic discrepancy between GWAS and reference samples increased (Supplementary Fig. 8). This observation demonstrates the importance of choosing a reference

sample that is genetically as close to the GWAS sample as possible in the analysis of summary data[31].

The parameter estimates were largely consistent between SBayesS and SBayesRS except for polygenicity, of which the estimate from SBayesRS was higher than that from SBayesS (Supplementary Fig. 9a). This is because, on one hand, SBayesS has a relatively low power to detect SNPs with very small effect sizes due to its assumption of a single normal distribution; on the other hand, SBayesRS tends to overestimate the number of SNPs with very small effect sizes due to the insufficient power to distinguish very small effect sizes from zero, as suggested by simulation (Supplementary Fig. 9c). Nevertheless, the number of SNPs with relatively large effects estimated from SBayesS was mostly consistent with that from SBayesRS (Supplementary Fig. 9b).

Finally, we tested the method in application to ascertained case-control data by simulation. The parameter estimates were nearly unbiased regardless of whether cases were oversampled, although the sampling variances of the estimates of polygenicity and S were relatively large in some simulation scenarios where the number of cases was relatively small (Supplementary Fig. 10).

**Analyses of GWAS summary data from the UKB and other published studies.** We applied SBayesS to analyse the full release of the UKB data, including 26 complex traits and 9 common diseases (Supplementary Table 1). Although individual-level data are available in the UKB, application of the standard BayesS to ~350k unrelated individuals with ~1.1 million HapMap3 SNPs is computationally prohibitive. Prior to running SBayesS, we carried out standard quality control (QC) of the data (see the "Methods" section) and used linear regression to perform a GWAS analysis in unrelated individuals to generate summary statistics for each trait. We also applied SBayesS to data for 9 other complex common diseases from published GWAS of very large sample size where only summary statistics are available (Supplementary Table 2). In the analysis of the UKB data, we used the sparse LD matrix computed from a random sample of 50k unrelated individuals. For the analysis of data from published GWAS of which nearly all the samples are of European ancestry, the GERA[32] sample was used as the LD reference. To mitigate the problem due to inconsistent LD between the GWAS and reference samples, we excluded SNPs in the major histocompatibility complex

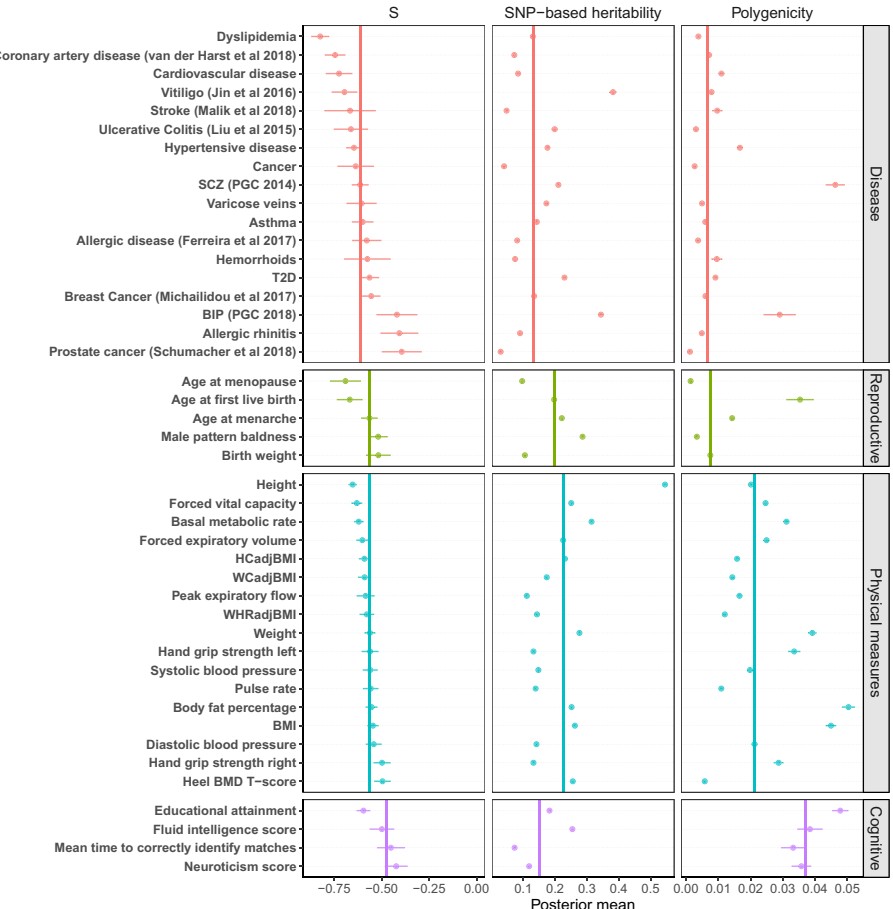

**Fig. 2 Estimation of the three genetic architecture parameters for 35 traits from the UKB and 9 common diseases from published GWAS.** Shown are the posterior means (dots) and standard errors (horizontal bars) of the parameters for each trait. The colour indicates the UKB trait category that the trait belongs to. The vertical bar shows the median of the estimates across traits in each category.

(MHC) region although the SBayesS results with and without the MHC region were very similar (Supplementary Fig. 11). The SNP-based heritability estimates for the diseases were converted to those on the liability scale[33].

On average across the 44 complex traits (including diseases), 1.8% of the 1.1 million common HapMap3 SNPs explained 18% of the phenotypic variance (Fig. 2 and Supplementary Tables 1, 2). The estimate of $h^2_{SNP}$ for height was 0.545 (posterior standard error or p.s.e. = 0.003), consistent with those in previous studies using different approaches and data sets[6,14,34–36]. The most polygenic traits (i.e., body fat percentage, educational attainment and schizophrenia) had about 5% (~55,000) SNPs with nonzero effects. The least polygenic traits were prostate cancer, age at menopause and male pattern baldness, which were affected by about 0.1–0.3% (1000–3000) common SNPs. The estimate of $S$ was significantly negative ($P < 0.001$) in all the traits analysed (median $\hat{S} = −0.578$, SD = 0.096), suggesting a pervasive action of negative selection on the trait-associated variants. We also re-ran the analysis for the 9 public GWAS data sets with the UKB subsample as the LD reference and found that the results were highly consistent with those using LD from the GERA (Supplementary Fig. 12).

We used the UKB classification code to classify the 44 traits into four categories related to disease, reproduction, physical measures, and cognition (Supplementary Table 3). The estimates of the genetic architecture parameters varied across traits and appeared to have distinct patterns in different categories (Fig. 2). Physical measures had the highest median SNP-based heritability

(0.225), followed by reproductive traits (0.197). The median polygenicity estimate was the lowest for diseases (0.007) and reproductive traits (0.008) and the highest for cognitive traits (0.037). The estimates of polygenicity for psychiatric disorders such as schizophrenia ($\hat{\pi} = 0.046$, p.s.e. = 0.003) and bipolar disorder ($\hat{\pi} = 0.034$, p.s.e. = 0.009) were substantially higher than that for other types of disease and comparable to those for the cognitive traits, consistent with the high polygenicity for brain-related traits reported in previous studies[11,28]. The absolute median value of $\hat{S}$ was the highest for diseases, especially cardiovascular diseases, and the lowest for cognitive traits, with a relatively large variability in $\hat{S}$ for diseases. We observed similar results from SBayesRS but with higher $\hat{\pi}$ (Supplementary Fig. 13), in line with the observations from the benchmark analysis above.

To investigate the diversity of genetic architecture in more traits, we applied SBayesS to GWAS summary data from the Neale Lab (http://www.nealelab.is/uk-biobank) for 274 UKB traits, among which 130 passed the convergence test and 110 of these were not included in the analyses above (Supplementary Table 4). The traits that failed to converge tended to have much smaller sample size or $\hat{h}^2_{SNP}$ compared to the ones that converged (mean $n = 231$k and $\hat{h}^2_{SNP} = 0.223$ for the converged vs. mean $n = 73$k and $\hat{h}^2_{SNP} = 0.044$ for the non-converged), but we did not filter traits by $\hat{h}^2_{SNP}$ to avoid direct ascertainment bias. Figure 3 shows the distributions of the estimated genetic architecture parameters for the total 155 traits (including 18 common diseases) from the UKB and published GWAS. Similar to that

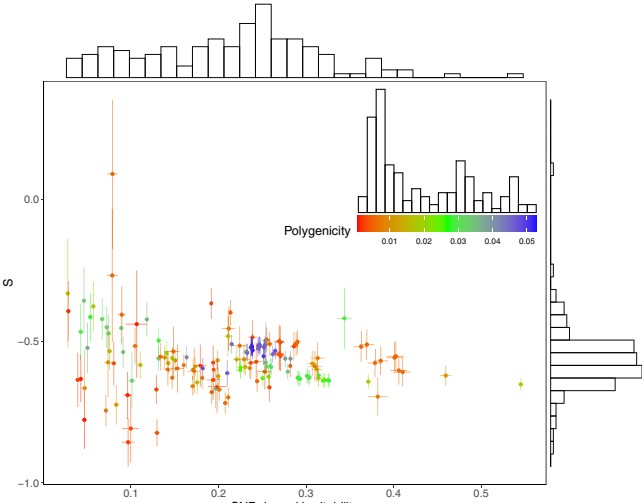

**Fig. 3 Estimation of the genetic architecture parameters for 155 complex traits.** Shown are the results from the SBayesS analyses using summary data for 130 traits from the Neale Lab and 25 traits from our GWAS analyses. The estimated $S$ is plotted against the estimated SNP-based heritability with the histograms showing the marginal distributions of the estimates. Data are presented as posterior means ± posterior standard errors. Colour indicates the estimate of polygenicity for each trait where the scale and distribution are shown in the inset graph.

for the 44 traits analysed above, the distribution of $\hat{S}$ appeared symmetric at about $-0.6$, with 79% of the estimates within the range of $-0.7$ to $-0.5$ (median $= -0.576$) across traits.

**Prediction of evolutionary parameters for complex traits.** Although a negative estimate of $S$ is a signature of negative selection, the numeric interpretation of $\hat{S}$ is still not clear. For example, our results showed that most traits had $\hat{S}$ at about $-0.6$; does it mean that negative selection acted on the associated variants with similar selection strength among traits? To answer this question, we performed forward simulations[26] to study the variational patterns of the genetic architecture parameters in a variety of evolutionary scenarios. We focused on three main evolutionary parameters: average selection coefficient ($\bar{s}$), proportion of mutational targets ($\pi_{\mathrm{m}}$, i.e., the proportion of DNA sequence at which mutations can affect the trait) and mutational heritability ($h_{\mathrm{m}}^2 = \sigma_{\mathrm{m}}^2/\sigma_{\mathrm{e}}^2$ with $\sigma_{\mathrm{m}}^2$ being the amount of additional additive genetic variance arising from new mutations in each generation and $\sigma_{\mathrm{e}}^2$ being the environmental variance $\sigma_{\mathrm{e}}^2$), and set a large range of values for each parameter (see the "Methods" section). The simulations were carried out based on a 100-Mb genome with a variable effective population size inferred from a demographic model[37]. The selection coefficients were sampled from either a normal distribution or a mixture distribution of many small and some very large values. In the last generation of selection, we used two pleiotropic models (the Simon et al.[12] and Eyre-Walker[3] model) to generate causal effects of the mutations on the focal trait (see the "Methods" section). Based on the LD correlation matrix computed from the unrelated individuals in the final generation of the simulated data, we directly simulated GWAS summary statistics with an equal sample size as in the UKB data analysis[11,38] (see the "Methods" section). The genetic architecture parameters $h_{\mathrm{SNP}}^2$, $\pi$ and $S$ were estimated by SBayesS and SBayesRS using 36k SNPs randomly sampled from the common sequence variants, a comparable SNP density to that of the 1.1 million HapMap3 common SNPs used in the real trait analysis ($1.1 \times 10^6 \times 1 \times 10^8/3 \times 10^9 = 36,000$).

Repeating the simulation with different values of $\bar{s}$, $\pi_{\mathrm{m}}$ and $h_{\mathrm{m}}^2$ produced a landscape of the genetic architecture under different scenarios (Fig. 4). Our results showed that $|\hat{S}|$, $\hat{\pi}$ and $\hat{h}_{\mathrm{SNP}}^2$ increased with the increasing levels of the corresponding evolutionary parameters $\bar{s}$, $\pi_{\mathrm{m}}$ and $h_{\mathrm{m}}^2$, respectively. The results were generally consistent regardless of the use of SNPs (36k common SNPs or the actual common causal variants), estimation method (SBayesS or SBayesRS), simulation model (the Simons et al. or Eyre-Walker model), or the underlying distribution of selection coefficients (mixture or normal distribution) (Fig. 4 and Supplementary Figs. 14–17). In addition to the direct impact of the evolutionary parameters on the corresponding genetic architecture parameters, $\bar{s}$ had negative effects on $\hat{h}_{\mathrm{SNP}}^2$ and $\hat{\pi}$ because of causal variants with large effect sizes were purged by negative selection. Using a linear regression analysis of the true causal effects, we obtained the ordinary least-squares (OLS) estimate of $S$ and used it as a proxy of the true value (see the "Methods" section). Regarding to the different distributions of selection coefficients, the true values of $S$ were not highly correlated (Supplementary Fig. 18; $r = 0.628$), suggesting the distribution of selection coefficient plays a role in determining $S$. Compared to the true value of $S$, the SBayesS estimate based on the causal variant genotypes tended to be more negative when selection strength was weak and selection coefficients followed a mixture distribution (Fig. 4). This is because SBayesS assumes normality of the SNP effects whereas the true distribution is a multi-component mixture, supported by our results of maximum-likelihood estimation assuming normality (Supplementary Note and Supplementary Fig. 19). Compared to SBayesS, SBayesRS is more robust to the distribution of causal effects in the estimation of the $S$ parameter (Fig. 4 and Supplementary Fig. 17).

Next, we used a polynomial regression model to associate the evolutionary parameters ($\bar{s}$, $\pi_{\mathrm{m}}$ and $h_{\mathrm{m}}^2$) with $\hat{h}_{\mathrm{SNP}}^2$, $\hat{\pi}$ and $\hat{S}$ in the entire simulation dataset, and leveraged this association to predict the evolutionary parameters in real data (see the "Methods" section). We demonstrated by a cross-validation analysis in the simulated data that the three evolutionary parameters can be predicted with reasonably high accuracy (Supplementary Fig. 20). We then applied this prediction model to the 44 traits analysed above. Overall, the real trait prediction results were robust to the evolutionary model used for training data simulation and the statistical method for genetic architecture parameter estimation (Fig. 5). Here, we focus on the results from the mixture model for selection coefficients, as similar results were observed from the model assuming normality (Supplementary Fig. 21). The predicted $\bar{s}$ were mostly between $10^{-4}$ and $10^{-3}$ with a mean of 0.0007 across traits (Fig. 5), in line with the estimates from prior work[12]. Cancer, stroke, and coronary artery disease showed the highest predicted $\bar{s}$ (Supplementary Fig. 22), albeit the per-trait $\bar{s}$ had a wide confidence interval which covered a range of possible values by a factor of 10. The predicted $\pi_{\mathrm{m}}$ was ~1% on average across traits, meaning that ~30 million base pairs of the human genome were mutational targets for a complex trait. The mean predicted $h_{\mathrm{m}}^2$ was 0.001 with relatively small variability across traits comparing to the other two parameters, which can be regarded as a justification of our projection approach because $h_{\mathrm{m}}^2$ is a rather conservative parameter with estimates of all ~0.001 across traits even in different species[39].

While the predicted $h_m^2$ was similar across traits, the predicted $\bar{s}$ and $\pi_{\mathrm{m}}$ were significantly different between some trait categories (Fig. 5 and Supplementary Fig. 22). Common diseases had a mean $\bar{s}$ of 0.0010, which was significantly higher than that of 0.0005 for physical measures (median $P$ value $= 0.015$ among the four groups of estimation methods and pleiotropic models). Compared to physical measures (mean $\pi_{\mathrm{m}} = 0.011$), the mean $\pi_{\mathrm{m}}$ was significantly lower for disease (0.005, median $P = 0.003$) and

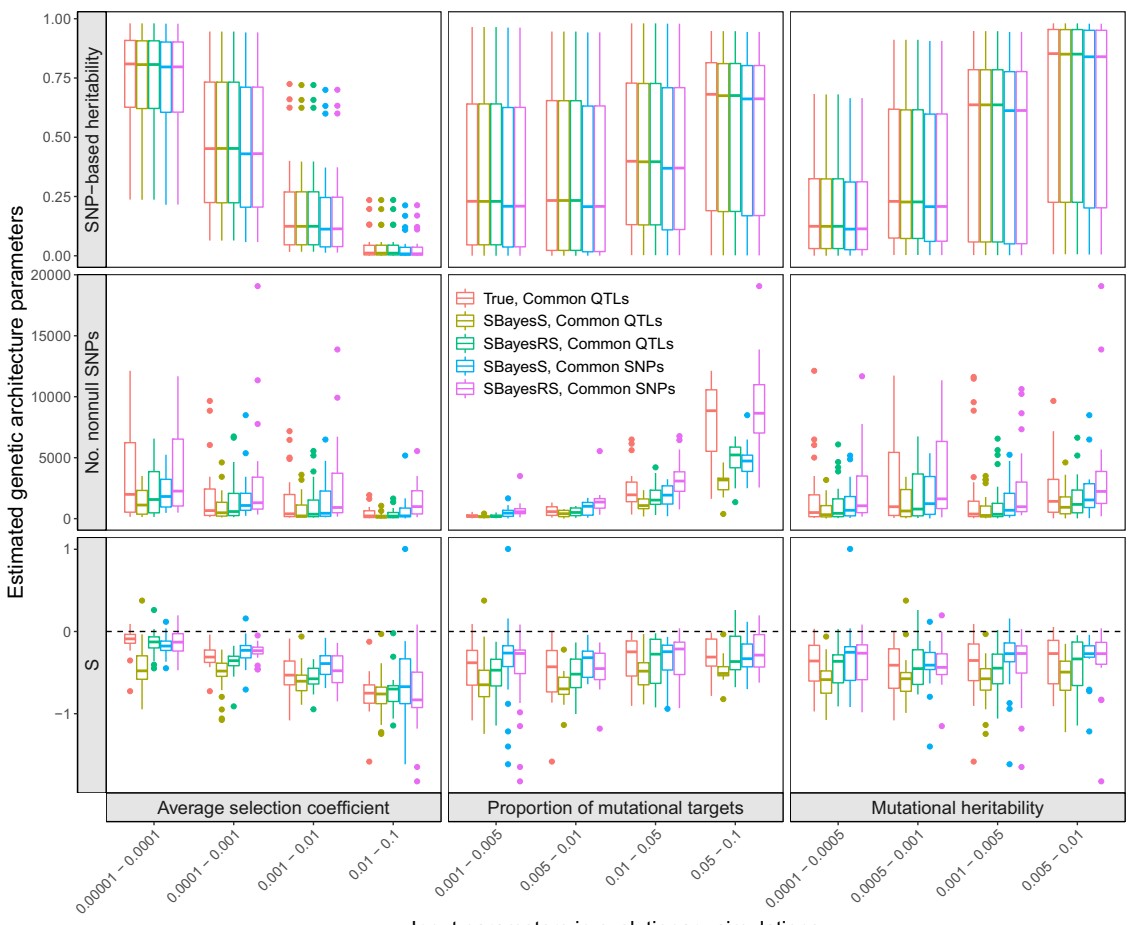

**Fig. 4 Variational patterns of the estimated genetic architecture parameters under different scenarios of evolutionary simulations.** The selection coefficients followed a mixture distribution, and the Simons et al. pleiotropic model with $n_t = 1$ was used to generate genetic effects (see the "Methods" section). The x-axis shows the values of three input parameters in the evolutionary simulations. The y-axis shows the distribution of the genetic architecture parameter estimates, where the polygenicity parameter is represented by the number of nonnull SNPs for better benchmarking. Colours indicate the following methods: "True, Common QTLs"—parameters computed directly from the simulated genetic effects of all common causal variants; "SBayesS, Common QTLs" (or "SBayesRS, Common QTLs")—SBayesS (or SBayesRS) estimates using the genotype data of the common causal variants, "SBayesS, Common SNPs" (or "SBayesRS, Common SNPs")—SBayesS (or SBayesRS) estimates using the genotype data of 36k common SNPs. Each box plot shows the results of 25 independent simulation replicates. The band inside the box is the median, the bottom and top of the box are the first and third quartiles, respectively (Q1 and Q3), and the lower and upper whiskers are Q1−1.5 × IQR and Q3 + 1.5 × IQR, respectively, where IQR = Q3−Q1.

was significantly higher for cognitive traits (0.023, median $P = 0.017$).

**Analyses incorporating functional genomic annotations.** The functional annotation categories used in our analysis were from the LDSC baseline model[15]. We excluded continuous annotations and annotations with flanking windows, resulting in 21 annotation categories such as the coding, regulatory, repressed and conserved regions (Supplementary Table 5). We applied SBayesS-strat to the 35 UKB traits (including 9 diseases), and combined the parameter estimates across traits for each functional category (see the "Methods" section). Considering the extensive overlaps between annotation categories (Supplementary Fig. 23), we ran SbayesS-strat analysis with a two-component model (SNPs in an annotation category versus the other SNPs) and computed the enrichment of each of the genetic architecture parameters using the SNPs in the focal annotation category in comparison to the genome-wide estimate using all SNPs. The fold enrichment of per-SNP heritability was correlated with that of polygenicity across annotation categories ($r = 0.762$; Fig. 6a). The per-SNP heritability and polygenicity were

enriched in functionally important categories, such as transcription start sites (TSS), 3'- and 5'-UTRs, and conserved, enhancer and coding regions, but depleted in repressed regions. This result suggests that a functional category that explains a greater fraction of heritability tends to have a larger number of nonnull variants, consistent with the findings from a recent study[11]. However, for some categories, such as coding and conserved regions, the fold enrichment of per-SNP heritability was greater than that of polygenicity, suggesting an enrichment of larger effect sizes in these regions. To distinguish between the contributions of the number and the magnitude of the nonzero effects to $h_{SNP}^2$, we estimated per-NZE heritability (per-nonzero effect heritability $h_{NZE}^2(c) = \frac{h_{SNP}^2(c)}{m_{NZ}(c)}$ where $m_{NZ}(c)$ is the number of SNPs with nonzero effects in category $c$. While the fold enrichment of $h_{NZE}^2(c)$ was close to or smaller than one in most categories, the enrichment was the largest in the coding and conserved regions (Fig. 6b), suggesting that the enrichment of per-SNP heritability in these categories was not only because of the larger number of nonnull variants but also the larger effect sizes, confirmed by simulations (Supplementary Fig. 24). The median value of $\hat{S}$ was $-0.540$, ranging from $-0.739$ (s.e.m. =

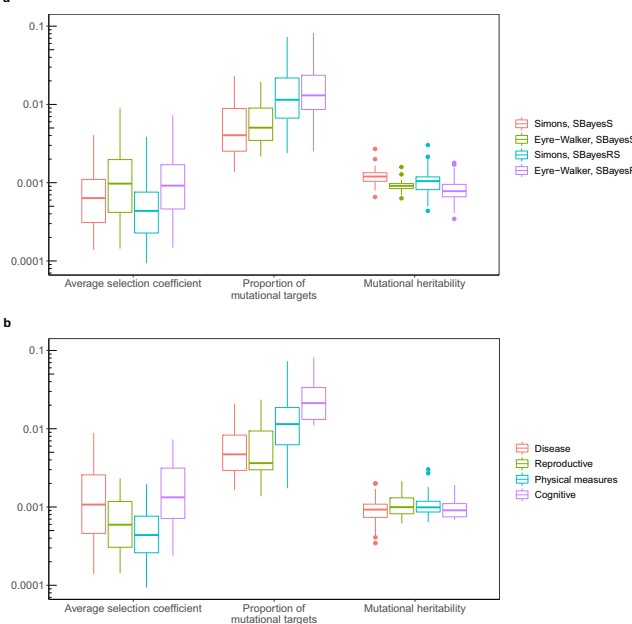

**Fig. 5 Prediction of the evolutionary parameters for 44 complex traits and diseases based on a negative selection model where selection coefficients followed a mixture distribution. a** Distribution of the predicted evolutionary parameters under different scenarios: methods used for estimating the genetic architecture parameters (SBayesS and SBayesRS) and pleiotropic effect models used for simulations (the Simons et al. and Eyre-Walker model), shown by colours. Each box plot shows the results for 44 complex traits. **b** Distribution of predicted evolutionary parameters for four trait categories, shown by colours. Each box plot shows the results for a number of traits in a category, with each trait having four results from analyses using different estimation methods and simulation models.

0.041) in coding regions to −0.361 (s.e.m. = 0.066) in TSS (Fig. 6c). The negative $\hat{S}$ in all functional categories suggests a widespread negative selection across the whole genome, and the largest $|\hat{S}|$ in the coding regions among all the functional categories highlighted the action of negative selection in the biologically most important regions.

Our estimates of per-SNP heritability enrichment were consistent with those from S-LDSC[15,40,41] for most annotation categories (Supplementary Fig. 25). However, S-LDSC reported a much larger enrichment for the conserved region category, followed by the coding region category. This may be due to the different assumptions made in the two methods, i.e., SBayesS-strat assumes a sparse genetic architecture whereas S-LDSC does not explicitly assume a mixture model, as both the coding and conserved regions categories were enriched for the number of nonzero effects and the magnitude of effect sizes (Fig. 6b). Another explanation could be that the SBayesS-strat estimate is from a separate analysis of a focal category at a time conditioning on all the other SNPs with no overlap among categories whereas the S-LDSC estimates are from a joint analysis of all the categories with overlaps.

## Discussion

We have developed an efficient summary-data-based method to estimate the joint distribution of effect sizes and MAF as well as SNP-based heritability, polygenicity and joint SNP effects. By analysing GWAS summary statistics from the public domain, we detected pervasive signatures of negative selection in the genetic

architecture for a wide range of complex traits including common diseases (Figs. 2 and 3). Our results support a model of negative selection, that is, most new nonneural mutations are deleterious to fitness such that mutations with larger effects on fitness are more likely to be eliminated or kept at lower frequencies in the population by selection.

Most traits had $\hat{S}$ at about −0.6 with diverse estimates of $h^2_{SNP}$ and polygenicity, implying that the model with $S = −1$ originally used in the GREML method[42] is more appropriate than the model with $S = 0$ for most complex traits. Schoech et al.[7] linked the $S$ parameter (denoted by $\alpha$ in their model) to the $\tau$ parameter in Eyre-Walker's model[3] and further drew inference on the average genome-wide selection coefficient. However, our forward simulations have shown that inference regarding the strength of selection cannot be made based solely on $S$ but should take into account other genetic architecture parameters as well as the distribution of effect sizes. Despite the narrowly distributed $\hat{S}$ across traits, the predicted strength of selection per trait can vary by orders of magnitude (Fig. 5 and Supplementary Fig. 22). By extrapolating our results based on HapMap3 common SNPs, we estimate that, on average, ~1% of human genome sequence are mutational targets for a complex trait with an average selection coefficient of 0.0007, giving rise to additional additive genetic variance of 0.001 (in the unit of environmental variance) in each generation. The large estimates of mutational target size per trait implicate widespread pleiotropy across the genome, consistent with the result of a recent study that 90% of GWAS loci affect multiple traits[43]. Our results suggest a relatively small mutational target size but relatively strong selection on variants for common diseases and relatively large mutational target size for cognitive traits, in line with the previous finding that brain-related traits are highly polygenic and the associated genetic variants are likely under strong selection[11].

Our polygenicity parameter $\pi$ represents the proportion of SNPs with nonzero effects; this definition has also been used previously[13,28,35,44–47]. Our forward simulations showed that $\pi$ is driven by both the mutational target size and selection strength, with increased average selection coefficient resulting in decreased $\hat{\pi}$. This is because negative selection removed causal variants of large effects as well as SNPs in LD with them (a phenomenon known as background selection). O'Connor et al.[11] proposed an alternative definition of polygenicity, the effective number of independently associated variants or $M_e$, which accounts for the distribution of per-SNP heritability across the genome. Despite the difference in definition, both $\pi$ and $M_e$ estimates varied with the number of causal variants under different scenarios (Supplementary Fig. 26). In addition, the estimates of $\pi$ and $M_e$ were highly correlated ($r = 0.876$) with a regression slope of $\hat{\pi}$ on $M_e$ estimates = 3.4. This is highly consistent with the result reported in O'Connor et al. that their $M_e$ estimates were highly correlated with the estimates from our previous study[13] for a number of traits ($r = 0.9$) but 4× smaller (Fig. S4 in O'Connor et al.).

Since we only detected signatures of negative selection in real traits, our evolutionary simulations focused on the models of negative selection. To investigate the impact of both negative and positive selections, we extended our simulation scenarios by considering two additional positive selection-related parameters: average positive selection coefficient and proportion of beneficial mutational targets (see the "Methods" section). When considering both negative and positive selections in the simulations, we observed more complicated relationships between the genetic architecture and evolutionary parameters (Supplementary Fig. 27), which, however, could still be used for prediction. Our results showed that the predicted $\bar{s}$, $\pi_m$ and $h^2_m$ were consistent with those predicted above considering negative selection only (Supplementary Fig. 28), except that the estimated strength of

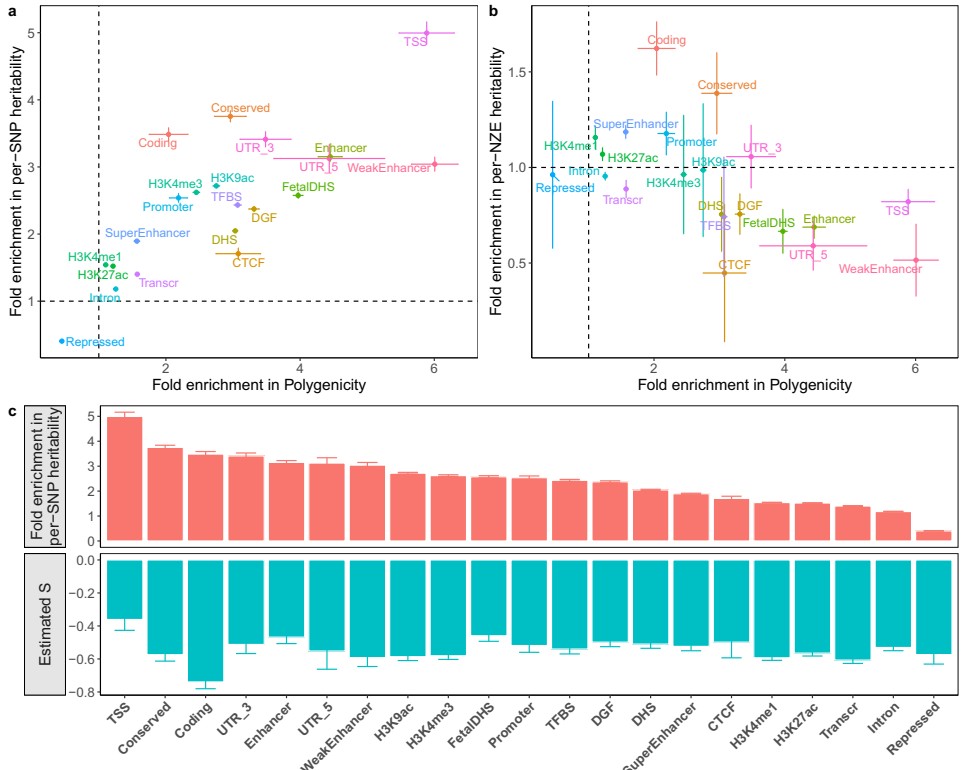

**Fig. 6 Characterisation of the genetic architecture in 21 functional genomic annotation categories using the two-component SBayesS-strat model. a** Fold enrichment of per-SNP heritability against that of polygenicity. **b** Fold enrichment of per-NZE (per-nonzero effect) heritability against that of polygenicity. Colours indicate different annotation categories. **c** Estimated $S$ (green) across annotation categories ranked by per-SNP heritability enrichment (red). Each dot or histogram is the median across 35 UKB traits (including diseases). Each bar indicates the standard error of the mean.

negative selection became weaker for cognitive traits, suggesting that positive selection may also play a role in shaping the genetic architecture of cognitive traits.

The biologically important categories, such as the TSS, conserved, UTR and coding regions, had the highest enrichment in per-SNP heritability, most of which also had the highest enrichment in polygenicity, whereas the repressed regions were depleted in both parameters (Fig. 6). The concordance in functional enrichment between the two parameters reflects an uneven distribution of the number of causal variants across functional categories, consistent with the finding from prior work[11]. We further observed enrichment of per-NZE heritability in conserved and coding regions, suggesting larger effect sizes of nonnull SNPs in these regions compared to genome average. It is of note that coding regions showed the largest $|\hat{S}|$ among all the functional categories (in line with Speed et al.[9]) with significant enrichment of per-NZE heritability, likely because of the coding mutations by nature having larger effect sizes and/or a mixture of negative selection on coding mutations with detrimental effects and positive selection on those with favourable effects. It is surprising to observe a relatively large $|\hat{S}|$ in repressed regions, which may be a consequence of overlaps between functional annotations. Another possible explanation could be that positive selection has suppressed the signature of negative selection in non-repressed (biologically important) regions, consistent with our observation that $S$ was closer to zero in the simulations considering both positive and negative selections than in the simulations only considering negative selection (Supplementary Fig. 27a versus Fig. 4).

There are several limitations in this study. First, our inference on negative selection is based on HapMap3 common SNPs and

therefore may not hold for the unobserved rare variants. In fact, we found by forward simulations a weaker magnitude of $S$ in rare variants because the very rare variants were mostly new mutations whose relationship between effect size and MAF had not yet been shaped by selection, which diluted the selection signals from the variants under selection (Supplementary Fig. 29). This suggests that the true $S$ parameter is allelic age dependent and subject to the combined effect of mutation, selection and genetic drift. An apparent change in the effect size–MAF relationship when moving toward low MAF was also reported by Schoech et al.[7]. Second, independence of chromosomes is assumed in our model. This may not hold if there was non-random mating in the ancestral population. For example, assortative mating would introduce positive correlations between trait-increasing alleles located on different chromosomes, and therefore increase heritability in the equilibrium population, e.g., for height[48]. Third, our definition of polygenicity is based on the number of SNPs with nonzero effects ($m_{NZ}$), which may not be an unbiased estimator of the number of causal variants ($m_C$) especially when the causal variants are not observed. For example, $m_{NZ}$ will tend to be smaller than $m_C$ if some causal variants are not well tagged by any SNP markers but tend to be larger than $m_C$ if they are in high multi-locus LD with a number of SNPs. Thus, our polygenicity estimate should be best used to compare traits using the same set of SNPs, rather than an unbiased estimate of the number of causal variants. Fourth, we did not attempt to predict the evolutionary parameters for functional genomic categories because it would require simulating a genome with functional partitioning. Despite these limitations, our study highlights the impact of negative selection on the genetic architecture across complex traits and in different functional genomic regions. In addition to a better understanding of the genetic

architecture, our methods can also be applied to genetic mapping and polygenic risk prediction through the use of the joint SNP effect estimates or the characterised underlying distributions of effect sizes as prior knowledge for other methods[49].

## Methods

**SBayesS**. Let us consider an individual-level data-based multiple regression model for a GWAS cohort:

$$\mathbf{y} = \mathbf{X}\boldsymbol{\beta} + \mathbf{e} \qquad (1)$$

where $\mathbf{y}$ is the vector of phenotypes adjusted for all fixed effects, $\mathbf{X}$ is the column-centred genotype matrix, $\boldsymbol{\beta}$ is the vector of SNP effects, and $\mathbf{e}$ is the vector of residuals with $Var(\mathbf{e}) = \mathbf{I}\sigma_e^2$ for a cohort of unrelated individuals. Assuming Hardy–Weinberg equilibrium (HWE), the variance of genotype dosage (0, 1, 2) of SNP $j$ is $h_j = 2p_jq_j$, where $p_j$ is MAF and $q_j = 1 - p_j$. Let $\mathbf{D}$ be a diagonal matrix with $D_{jj} = \mathbf{X}_j'\mathbf{X}_j = h_jn_j$, where $n_j$ is per-SNP sample size. Multiplying both sides of the equation by $\mathbf{D}^{-1}\mathbf{X}'$ gives

$$\mathbf{D}^{-1}\mathbf{X}'\mathbf{y} = \mathbf{D}^{-1}\mathbf{X}'\mathbf{X}\boldsymbol{\beta} + \mathbf{D}^{-1}\mathbf{X}'\mathbf{e}$$

Note that $\mathbf{D}^{-1}\mathbf{X}'\mathbf{y} = \mathbf{b}$, the vector of least-squares estimates of SNP marginal effects from GWAS, and $\mathbf{D}^{-1}\mathbf{X}'\mathbf{X} = \mathbf{D}^{-\frac{1}{2}}\mathbf{B}\mathbf{D}^{\frac{1}{2}}$, where $\mathbf{B} = \mathbf{D}^{-\frac{1}{2}}\mathbf{X}'\mathbf{X}\mathbf{D}^{-\frac{1}{2}}$ is the LD correlation matrix among all SNPs[50]. Let $\mathbf{W} = \mathbf{D}^{-\frac{1}{2}}\mathbf{B}\mathbf{D}^{\frac{1}{2}}$ and $\boldsymbol{\varepsilon} = \mathbf{D}^{-1}\mathbf{X}'\mathbf{e}$. Then, the above equation can be written as

$$\mathbf{b} = \mathbf{W}\boldsymbol{\beta} + \boldsymbol{\varepsilon} \qquad (2)$$

In contrast to the identity structure of residual variance in model (1), the residuals in model (2) are dependent of LD, as

$$Var(\boldsymbol{\varepsilon}) = \mathbf{D}^{-\frac{1}{2}}\mathbf{B}\mathbf{D}^{-\frac{1}{2}}\sigma_e^2 = \mathbf{R}\sigma_e^2 \qquad (3)$$

This is a generic form of summary-data-based Bayesian regressions, which is similar to Zhu and Stephens's RSS model[35]. As in BayesS, we assume the effect size is related to MAF through a parameter $S$:

$$\beta_j \sim N\left(0, h_j^S\sigma_\beta^2\right)\pi + \phi(1 - \pi) \qquad (4)$$

where $\phi$ is a point mass at zero, and $S$ (the relationship between MAF and effect size), $\sigma_\beta^2$ (the effect variance factor common to all SNPs) and $\pi$ (the proportion of SNPs with nonzero effects, i.e., the polygenicity) are considered as unknown, with prior distributions of a standard normal, a scaled inverse chi-squared distribution (Supplementary Note), and a uniform distribution between zero and one, respectively. Specifying a different prior distribution to $\beta_j$ gives a form of other summary-data-based Bayesian alphabet models[29].

When the LD correlations are computed using all SNPs in the GWAS sample, models (1) and (2) are equivalent in terms of posterior inference because the GWAS estimates of SNP effects ($\mathbf{b}$) and LD correlation matrix ($\mathbf{B}$) are sufficient statistics for the joint posterior distribution of $\boldsymbol{\beta}$ (Supplementary Note). Compared to model (1), model (2) allows us to incorporate LD information from a different reference sample from the GWAS sample for which the individual-level data are often not accessible. Further, it is often not practical to compute and store the entire LD matrix in the memory. Therefore, we used a sparse LD matrix that ignores the small LD correlation estimates due to sampling variation, but still accounted for the sampling variance of LD correlation in the model (Supplementary Note). In our GCTB software[13] where SBayesS is implemented, we have developed a parallel computing strategy to facilitate the computation of the LD matrix. Once the LD matrix is computed, it can be used repeatedly in the GWAS summary-data analysis for different traits.

**MCMC and convergence**. We used MCMC algorithm to generate 50,000 posterior samples (the first 20,000 discarded as burn-in) from the joint posterior distribution of model parameters, based on which statistical inference was made. Details of the MCMC sampling scheme are shown in the Supplementary Note. The posterior mean was used as the point estimator, with the statistical uncertainty quantified by the posterior variance or its square root (posterior standard error), as shown below. We ran four parallel chains with different starting values of the parameters randomly sampled from their prior distributions. Following the method proposed by Gelman and Rubin[27], we estimated the posterior variance by

$$\widehat{Var}(\theta|y) = \frac{T-1}{T}W + \frac{1}{T}B$$

where $T$ is the chain length, $W$ is the within-chain variance, and $B$ is the between-chain variance.

To assess convergence in MCMC, we computed the potential scale reduction statistic

$$\hat{R} = \sqrt{\frac{\widehat{Var}(\theta|y)}{W}}$$

for each of the model parameters. As suggested by Gelman and Rubin, $\hat{R} < 1.2$

generally indicates good convergence. Thus, we concluded convergence for a trait analysis when all the three genetic architecture parameters had $\hat{R} < 1.2$.

**Sparse LD matrix**. For computational efficiency, we used a sparse LD matrix in the analysis where LD due to sampling variation were set to be zero. To this end, we tested whether LD between each pair of SNPs on the same chromosome is zero in the population when computing the LD correlation matrix using a reference sample. Under the null hypothesis that the true LD in the population is zero, we assume[51]

$$\frac{\tilde{B}_{jk}^2}{Var\left(\tilde{B}_{jk}^2\right)} = \tilde{n}_{jk}\tilde{B}_{jk}^2 \sim \chi_1^2$$

(tilde denotes quantities computed from the reference sample) and reject the null if the chi-squared statistic > 10 (corresponding to $P < 0.0016$). This is equivalent to a $r^2$ threshold of $2 \times 10^{-4}$ given a sample size of 50,000, resulting in each SNP, on average, being in LD with ~1000 SNPs on the same chromosome. We set $\tilde{B}_{jk}$ to be zero if the null hypothesis is not rejected or if the two SNPs are on different chromosomes, leading to a sparse LD matrix. The chi-squared threshold of 10 is chosen in order to balance the type I and II error rates. If a type I error occurs, i.e., the true correlation $\rho_{jk} = 0$ but $\tilde{B}_{jk}$ is not set to be zero, then as shown in the Supplementary Information, $s_{jk}^2 = \frac{\tilde{n}_{jk}+n_{jk}}{\tilde{n}_{jk}n_{jk}}\left(1 - \tilde{B}_{jk}^2\right)^2$, which is very likely to be larger than the true sampling variance $1/n_{jk}$. This would inflate the residual variance and therefore deflate the heritability estimate. If a type II error occurs, i.e., $\rho_{jk} \neq 0$ but $\tilde{B}_{jk}$ is set to be zero, then $s_{jk}^2 = \frac{1}{n_{jk}}$, which is very likely to be larger than the true sampling variance $\frac{\tilde{n}_{jk}+n_{jk}}{\tilde{n}_{jk}n_{jk}}\left(1 - \rho_{jk}^2\right)^2$. This would deflate the residual variance and therefore inflate the heritability estimate. Since the consequence of type II errors is worse, we use a not-too-stringent threshold to eliminate the LD due to sampling. This also suggests that LD reference sample size cannot be too small, otherwise, type II error rate would increase due to the loss of power. Since we only include non-zero elements in the LD matrix, it is faster by folds to run the summary-level data analysis with substantially less amount of memory needed.

**SNP-based heritability estimation**. In BayesS[13], we computed the genetic variance $\sigma_g^2$ as the variance of genetic values across individuals given the sampled values of $\boldsymbol{\beta}$ in each MCMC iteration. As described in Zhu and Stephens[35], this is equivalent to the following quadratic term of $\boldsymbol{\beta}$ given the LD correlation matrix:

$$\sigma_g^2 = \frac{\sum_{i=1}^{n}\left(\mathbf{X}_i\boldsymbol{\beta}\right)^2}{n} = \frac{trace\left[\left(\mathbf{X}\boldsymbol{\beta}\right)\left(\mathbf{X}\boldsymbol{\beta}\right)'\right]}{n} = \boldsymbol{\beta}'\frac{\mathbf{X}'\mathbf{X}}{n}\boldsymbol{\beta} = \boldsymbol{\beta}'\mathbf{B}\boldsymbol{\beta}$$

Given the right-hand-side updating strategy in MCMC (Supplementary Note), this quadratic term can be computed efficiently as the difference of two vector-by-vector products:

$$\boldsymbol{\beta}'\mathbf{B}\boldsymbol{\beta} = \boldsymbol{\beta}'\mathbf{r} - \boldsymbol{\beta}'\mathbf{r}_{adj}$$

where $\mathbf{r} = \mathbf{D}\mathbf{b}$ and $\mathbf{r}_{adj}$ is the adjusted $\mathbf{r}$ for $\boldsymbol{\beta}$. The residual variance ($\sigma_e^2$) is sampled from a scaled inverse chi-squared distribution with the mean mainly driven by

$$\frac{\mathbf{e}'\mathbf{e}}{n} = \frac{\mathbf{y}'\mathbf{y} - \boldsymbol{\beta}'\mathbf{r} - \boldsymbol{\beta}'\mathbf{r}_{adj}}{n}$$

where $\mathbf{y}'\mathbf{y}$ is estimated by the median value of $D_{jj}\left(n_j\text{SE}_j^2 + b_j^2\right)$ across SNPs, where $\text{SE}_j$ is the standard error of $b_j$. Conditional on $\sigma_g^2$ and $\sigma_e^2$, we computed $h_{\text{SNP}}^2 = \frac{\sigma_g^2}{\sigma_g^2 + \sigma_e^2}$ in each MCMC iteration, and used the mean over MCMC samples as the point estimator of the SNP-based heritability.

**SBayesRS**. Following the recently published SBayesR[29] model which assumes a mixture of a point mass at zero and multiple normal distributions with different variances, we extended SBayesS to this flexible multi-component mixture model to account for a more complex genetic architecture with a spectrum of very small to very large effect sizes. For each SNP effect, we assume

$$\beta_j \sim \sum_{k=1}^{4}\pi_k N\left(0, \gamma_k h_j^S\sigma_\beta^2\right)$$

where $\gamma_k = 0, 0.01, 0.1, 1$ for $k = 1, 2, 3, 4$, representing four mixture components of zero, small, medium and large effect size, respectively, with $\sum_{k=1}^{4}\pi_k = 1$. The priors for $S$ and $\sigma_\beta^2$ are as defined in SBayesS. The mixing probabilities $\boldsymbol{\pi}$ are assumed to have a Dirichlet distribution with hyperparameters set to one. The polygenicity is defined as the sum of fraction of SNPs in each nonnull component, i.e., $\pi = \pi_2 + \pi_3 + \pi_4$. The sparse LD matrix above or the shrunk LD matrix[29,35] used in the SBayesR study[29] can be used for SBayesRS analysis.

**SBayesS-strat**. SBayesS-strat is a two-component SBayesS model that allows the distributions of SNP effects in the focal annotation category, e.g., coding, regulatory and conserved regions, to be different from that in the rest of the genome.

Compared to other methods utilising functional annotations, such as S-LDSC[52], BayesRC[53] and RSS-E[54], a unique feature of the annotation-stratified SBayesS (referred to as SBayesS-strat) is that it allows the estimation of S in a specific functional annotation category. Compared to a recently published method, BLD-LDAK-Alpha[9], that estimates the S parameter (denoted by $\alpha$ in their model) based on an infinitesimal model, our method accounts for a sparse genetic architecture. In addition to the estimation of per-SNP heritability, polygenicity and S for each category, we also defined per-nonzero-effect (per-NZE) heritability ($h_{NZE}^2$) as the total heritability explained in a category divided by the number of nonzero effects in the category, which is helpful to understand whether the heritability enrichment is due to the larger number of associated variants or the larger magnitude of effect size compared to genome average. In addition to the category-specific parameters, we estimated the global parameters S, $\pi$, $h_{SNP}^2$ and $h_{NZE}^2$ empirically conditional on the sampled value of $\beta$ in each iteration of MCMC. The fold of enrichment of each parameter for each trait was then computed as $E_t\left[\theta_y^t/\theta^t\right]$ over T MCMC iterations. The estimation variation of the enrichment fold was quantified by the posterior variance as described above.

**Meta-analysis across traits**. We combined the SBayesS-strat estimates across traits by calculating the median fold enrichment for each functional category. We reported the median instead of the mean in order to minimise the impact of outliers, especially for the per-NZE heritability estimate for which the denominator (i.e., the number of nonzero effects in an annotation category) is often estimated with large sampling variance. To account for the phenotypic correlation among the traits, we estimated the effective number of traits ($n_e$) by performing an eigen decomposition on the phenotypic correlation matrix[55]:

$$n_e = \frac{\left(\sum_i \lambda_i\right)^2}{\sum_i \lambda_i^2}$$

where $\lambda_i$ is the $i$th eigenvalue of the phenotypic correlation matrix. Then, the posterior standard error of the mean was computed as

$$\text{s.e.m.} = \frac{\widehat{SD}(\theta|y)}{\sqrt{n_e}}$$

with $\widehat{SD}(\theta|y)$ being the standard deviation of the estimate across traits.

**GWAS summary statistics**. We performed GWAS analyses for 26 quantitative traits and 9 common diseases in the full release of the UKB data using PLINK 1.90 beta[56]. We used 348,501 unrelated individuals of European ancestry (estimated genetic relatedness from GCTA < 0.05)[57] and the imputed data provided by the UKB team[16]. We filtered HapMap3 SNPs[30] with MAF < 0.01, HWE test P value < $1 \times 10^{-6}$, missing genotype rate > 0.05, or imputation info score < 0.3. We further excluded SNPs in the Human Major Histocompatibility Complex (MHC) region, resulting in a total of 1,124,198 common SNPs for the analysis. The LD correlations in the reference samples were computed based on the effect alleles in the GWAS summary data. For quantitative traits, we standardised phenotypes to mean zero and variance one after removing the outliers (phenotype > 7 SD) and performed rank-based inverse normal transformation (RINT) within each sex group. Prior to GWAS, we pre-adjusted phenotypes by age, sex and first 10 principal components (PCs) provided by the UKB team after RINT if applied. For the publicly available summary statistics, we downloaded the data and matched the SNPs with those in the UKB data after excluding the strand ambiguous SNPs (i.e., A/T or C/G SNPs) in addition to the QC procedures above. For the GWAS summary data from the Neale Lab, we extracted 274 quantitative traits for which the GWAS was performed based on RINT phenotypes in their analysis pipeline.

**Evolutionary forward simulations**. We used SLiM3[26] to run evolutionary forward simulations. A large sequence of 100 Mb was simulated, where a proportion of new mutations ($\pi_m$) that had pleiotropic effects on fitness and trait emerged at random with an average selection coefficient of $\bar{s}$ and a mutational heritability of $h_m^2$ in each generation. The values of the three input parameters were sampled from uniform distributions at log10 scale: $\log_{10}(s) \sim U(10^{-5}, 10^{-2})$, $\log_{10}(\pi_m) \sim U(10^{-3}, 10^{-1})$ and $\log_{10}(h_m^2) \sim U(10^{-4}, 10^{-2})$. The mutation rate ($\mu_m$) was set to $1.65 \times 10^{-8}$ per base pair per individual per generation[58], and the recombination rate was set to $1 \times 10^{-8}$. We assumed a model of negative selection (see below for a scenario with positive selection), where all trait variants were deleterious to fitness and the other mutations were neutral. The causal effects were assumed to have a mixture of three normal distributions with small, medium and large variances:

$\beta_j \sim 0.7N\left(0, 0.01\sigma_\beta^2\right) + 0.25N\left(0, 0.1\sigma_\beta^2\right) + 0.05N\left(0, \sigma_\beta^2\right)$, where the mixing proportions were set to the average SBayesRS estimates across 44 real traits. Thus, the marginal variance of causal effects is

$V_\beta = \text{Var}\left(\beta_j\right) = (0.7 * 0.01 + 0.25 * 0.1 + 0.05)\sigma_\beta^2$. Given the input parameters $h_m^2$ and $\pi_m$, the mixture distribution parameter $\sigma_\beta^2$ can be computed because, according to population genetics theory[10], $h_m^2 = 2L\pi_m\mu_m V_\beta$ in the unit of environmental variance $\sigma_e^2$, with L being the genome length (100 Mb). For each trait

mutation, the selection coefficient was modelled by $s_j = k\beta_j^2$, where $k = \bar{s}/V_\beta$ given the input parameter $\bar{s}$. Because $\beta_j$ followed a mixture distribution, $s_j$ also followed a mixture distribution with small and large selection coefficients. To break up the perfect proportionality between selection coefficients and squared effect sizes, two pleiotropic effect models were used to remodel the trait effect of causal variant conditional on its selection coefficient toward the end of the selection process. The first model is the Simons et al.[12] model which assumes the causal effect on the focal trait (denoted as 1) following the distribution $\beta_{j1} \sim N(0, k^{-1}s_j/n_t)$ with $n_t$ being the number of traits on which the variant has an effect. The second model is the Eyre-Walker's model[3]: $\beta_{j1} = \delta_j S_j^\tau \left(1 + \varepsilon_j\right)$ with $\delta_j = 1$ or $-1$ determined at random, $S_j = 4N_e s_j$, $N_e = 10,000$ being the effective population size and $\varepsilon_j \sim N(0, \sigma^2)$. In the Eyre-Walker's model, $\tau$ is the key parameter and $\sigma^2$ is a nuisance parameter. For each set of causal variants, we simulated causal effects based on either the Simons et al. model with $n_t = 1, 2, 4$ or 10 or the Eyre-Walker model with $\tau = 0.2, 0.5, 0.8, 1.0$ and $\sigma^2 = 0.1$.

A demographic model inferred by Gravel et al.[37] with population bottleneck and expansion was used to simulate a population that had undergone selection for 58,000 generations. The simulation started with an ancestral base population of $N_e = 7310$, which was expanded to 14,474 after 52,080 generations, a long period of neutral burn-in to allow the population reach mutation-drift equilibrium (~1.3 million years assuming 25 years per generation). In generation 55,960, 1861 individuals were split from the base population into a descendant population to mimic the out-of-African dispersal. In generation 57,080, the population size was further reduced to 1032 and then increased with an exponential rate of 0.0038 until generation 58,000, reaching a final population size of 34,039. In the last generation of selection, we obtained the genotypes of ~2000 unrelated individuals (genomic relationship < 0.05) and computed the LD correlation matrix for all common causal variants and a random sample of 36k common SNPs, a comparable density as the SNPs used in the real trait analysis ($1.1 \times 10^6 \times 1 \times 10^8/3 \times 10^9 = 36,000$). Given the LD matrix and causal effects, we directly simulated the GWAS summary statistics for all variants[11,38]: $\hat{\alpha} \sim N(\hat{\mathbf{B}}\beta, \frac{1}{N}\hat{\mathbf{B}})$, where $\hat{\mathbf{B}}$ is the LD matrix, $\beta$ is the simulated causal effects based on different evolutionary models (the Simon et al. or Eyre-Walker model), and $N = 350k$ is the sample size in the UKB dataset (see the Supplementary Note for more details). Using this approach, we were able to simulate a GWAS data set with comparable statistical power as in the real data analysis. We then used the same methods as those used in real trait analysis (i.e., SBayesS and SBayesRS) to estimate the SNP-based heritability, polygenicity and S at either common causal variants or the random set of 36,000 common SNPs (excluding the causal variants). The true value of SNP-based heritability was computed as $\sigma_g^2/(\sigma_g^2 + \sigma_e^2)$, where $\sigma_g^2$ is the genetic variance yielded from the simulation and $\sigma_e^2 = 1$. The true value for polygenicity was represented by the number of common causal variants in the last generation of selection. The true S parameter was estimated using a linear regression

$$\log\left(\beta_j^2\right) = \alpha_0 + \alpha_1 \log\left(2p_j q_j\right) + \epsilon_j$$

where the slope $\alpha_1$ is an OLS estimate of S according to the BayesS model, and the residuals $\epsilon$ are independent. When the distribution of causal effects is a mixture of multiple normal distributions, fitting a single intercept or multiple intercepts with respect to the mixture components has negligible impact on the OLS estimate of S (Supplementary Fig. 30). We performed the whole simulation process 100 times with a mixture or normal distribution for selection coefficients, respectively.

To incorporate positive selection, we specified two more input parameters, i.e., proportion of trait mutations being beneficial ($\pi_{m,b}$) and average selection coefficient for the beneficial alleles ($\bar{s}_b$). Similar as above, we sampled their values from uniform distributions at log10 scale: $\log_{10}(\bar{s}_b) \sim U(10^{-5}, 10^{-2})$ and $\log_{10}(\pi_{m,b}) \sim U(10^{-3}, 10^{-1})$. In this scenario, a new mutation can either be beneficial or deleterious with a selection coefficient as modelled above given the average positive or negative selection coefficient. Note that only the Simons et al. model was used to simulate GWAS data in this scenario as the Eyre-Walker model only fits in the context of negative selection.

**Inference on evolutionary parameters**. We used a polynomial regression model to predict an evolutionary parameter from the estimates of the genetic architecture parameters for complex traits. The forward simulation data under either the Simons et al.[12] or Eyre-Walker[3] model with various settings were used as a reference to estimate the associations between an evolutionary parameter ($\bar{s}$, $\pi_m$ and $h_m^2$ at log10 scale for a positive parameter space) and the genetic architecture parameters in their original units. We tested the performance of this method by a cross-validation analysis in the simulated data, with 80% of the sample as training and the rest as validation. As shown in Supplementary Fig. 20, the three evolutionary parameters, $\bar{s}$, $\pi_m$ and $h_m^2$, can be predicted with reasonably high accuracy, and the prediction reached the maximum when the degree of polynomial was about 3. Given this result, we chose the degree of polynomial of 3 in the real trait prediction analysis. We used the entire simulation data set to build a polynomial regression model to predict each of the evolutionary parameters by jointly modelling the SBayesS estimates of $h_{SNP}^2$, S and $\pi$ (or the SBayesRS estimates of $h_{SNP}^2$, S, $\pi_1$, $\pi_2$, $\pi_3$ and $\pi_4$). We then applied these polynomial equations to predict the evolutionary parameters in real data for 44 complex traits.

**Reporting summary**. Further information on research design is available in the Nature Research Reporting Summary linked to this article.

## Data availability

This study makes use of individual-level genotype and phenotype data from UK Biobank Resource (application number: 12505) as well as GWAS summary data and functional genomic annotation data from the public domain. UK Biobank: https://www.ukbiobank.ac.ukhttps://www.ukbiobank.ac.uk; GERA: https://www.ncbi.nlm.nih.gov/projects/gap/cgi-bin/study.cgi?study_id=phs000674.v2.p2; UKB GWAS summary data from the Neale Lab: http://www.nealelab.is/uk-biobank; baseline-LD annotations: https://data.broadinstitute.org/alkesgroup/LDSCORE; HapMap3: https://www.sanger.ac.uk/resources/downloads/human/hapmap3.html. Sparse LD matrix of ~1.1 million HapMap3 SNPs computed from 50,000 unrelated UKB individuals of European ancestry: https://cnsgenomics.com/software/gctb/#Download.

## Code availability

SBayesS, SBayesRS and SBayesS-strat have been implemented in the GCTB (genome-wide complex trait Bayesian analyses) software tool, freely available at http://cnsgenomics.com/software/gctb. Other software used in this study include PLINK 1.90 (https://www.cog-genomics.org/plink2), SLiM3 (https://messerlab.org/slim), S-LDSC (https://github.com/bulik/ldsc), and GCTA (https://cnsgenomics.com/software/gcta).

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

## Acknowledgements

We thank V. Hivert for helpful discussions. This research was supported by the Australian Research Council (DP160101343, DP160101056, and FT180100186), the Australian National Health and Medical Research Council (1107258, 1078901, 1078037, 1113400, and 1177268) and the Westlake Education Foundation. This study makes use of data from dbGaP (accession: phs000788) and UK Biobank Resource (application number: 12505). A full list of acknowledgements for these datasets can be found in the Supplementary Information.

## Author contributions

J.Y. and J.Z. conceived the study and designed the experiment. J.Z. derived the analytical methods, conducted all analyses, and developed the software with assistance and guidance from A.X., L.J., L.R.L.-J., Y.W., H.W., Z.Z., L.Y., K.E.K., M.E.G., N.R.W., P.M.V. and J.Y. J.Z. and J.Y. wrote the manuscript with the participation of all authors. All authors reviewed and approved the final manuscript.

## Competing interests

The authors declare no competing interests.
