## [Peer Review File · Nature Communications]

Reviewer #1 (Remarks to the Author):

The authors have worked hard to address all of my comments, and the manuscript is greatly improved. I don't think any new analyses are needed.

Minor comments

1. I remain confused about the M_e analyses, which seem to show the opposite of the effect that is identified in O'Connor et al. 2018. A few things are surprising; I don't think new analyses are needed, but please look carefully at the results to see if they can be resolved:
 - a. In the absence of selection, M_e should be dominated by the largest-effect SNPs, a small fraction of all SNPs. Is this the case?
 - b. M_e really ought to change when switching from per-allele to per-normalized-genotype units (the latter being correct), as the former is only affected by selection to the extent that it causes fixation/loss, while the latter is strongly affected by selection as mediated by allele frequencies. If the results didn't change at all when switching units, there must be something strange going on – please check!
 - c. The authors state that negative selection had an effect on the total number of SNPs. Are different results obtained if polygenicity is quantified as a fraction of SNPs?
2. The distinction between "causal SNPs" and "SNPs with nonzero effects" seems like it needs more explanation. Is the former the estimand and the latter the estimator?
3. Regarding uncertainty in evolutionary parameters, the equation in the rebuttal whose LHS is $\text{Var}(\hat{\theta})$ cannot be correct, as the LHS is a fixed value and the RHS is a random variable (a function of \hat{X}). I suspect this is a minor issue/type-o.

Reviewer #1 (Remarks to the Author):

The authors have worked hard to address all of my comments, and the manuscript is greatly improved. I don't think any new analyses are needed.

Re: We thank the reviewer for all the constructive comments, which have led to a substantial improvement in our manuscript.

Minor comments

1. I remain confused about the M_e analyses, which seem to show the opposite of the effect that is identified in O'Connor et al. 2018. A few things are surprising; I don't think new analyses are needed, but please look carefully at the results to see if they can be resolved:

a. In the absence of selection, M_e should be dominated by the largest-effect SNPs, a small fraction of all SNPs. Is this the case?

Re: We looked at the result of the simulation for which the average selection coefficient was closest to zero (1.1×10^{-5}). There was in total 35,619 common SNPs including 414 causal variants. The M_e estimate was 64.3 when all 414 causal variants are included in the calculation. It reduced to 52.1 when including the top 229 (55%) causal variants each explained $> 0.01\%$ heritability, 44.6 when including 96 (23%) causal variants each explained $> 0.1\%$ heritability, and 23.2 when including only 20 (5%) causal variants each explained $> 1\%$ heritability. This result clearly shows that M_e was indeed dominated by a small fraction of large-effect SNPs.

b. M_e really ought to change when switching from per-allele to per-normalized-genotype units (the latter being correct), as the former is only affected by selection to the extent that it causes fixation/loss, while the latter is strongly affected by selection as mediated by allele frequencies. If the results didn't change at all when switching units, there must be something strange going on – please check!

Re: In the example above, we indeed observed a change of value of M_e when switching from per-allele to per-normalized-genotype units for the SNP effects. When using per-allele effect sizes, M_e was 52.8, and when using per-normalized-genotype effect sizes, M_e was 64.3. These observations and the observations above do not affect the statements about M_e in the latest version of our manuscript (lines 378-385).

c. The authors state that negative selection had an effect on the total number of SNPs. Are different results obtained if polygenicity is quantified as a fraction of SNPs?

Re: The results of our simulation study were consistent regardless of using the number of SNPs with nonzero effects or the proportion of SNPs with nonzero effects. We used the number instead of proportion because the estimates were on the same scale when using all SNP markers or the causal variants only in the analysis. In fact, when we derived the polynomial model for predicting evolutionary parameters, we used the proportion of SNPs with nonzero effects to match the polygenicity estimate from the real trait analysis (which is defined based on the proportion). Thus, our inference from the real data analysis would not change at all.

2. The distinction between “causal SNPs” and “SNPs with nonzero effects” seems like it needs more explanation. Is the former the estimand and the latter the estimator?

Re: When the causal SNPs are observed, the number of SNPs with nonzero effects (m_{NZ}) is an estimator of the number of causal SNPs (m_C). When the causal variants are not observed, m_{NZ} may not reflect the actual value of m_C because 1) if some causal variants are not tagged by any SNP markers, m_{NZ} will tend to be smaller than m_C and 2) if some causal variants are in low LD with any individual SNP but in high multi-locus LD with a number of SNPs, m_{NZ} will tend to be larger than m_C . We have clarified this as a limitation in the previous manuscript. In the revised manuscript, we have made the statement clearer as: “Third, our definition of polygenicity is based on the number of SNPs with nonzero effects (m_{NZ}), which may not be an unbiased estimator of the number of causal variants (m_C) especially when the causal variants are not observed. For example, m_{NZ} will tend to be smaller than m_C if some causal variants are not well tagged by any SNP markers but tend to be larger than m_C if they are in high multi-locus LD with a number of SNPs. Thus, our polygenicity estimate should be best used to compare traits using the same set of SNPs, rather than an unbiased estimate of the number of causal variants.”

3. Regarding uncertainty in evolutionary parameters, the equation in the rebuttal whose LHS is $Var(\hat{\theta})$ cannot be correct, as the LHS is a fixed value and the RHS is a random variable (a function of \hat{X}). I suspect this is a minor issue/type-o.

Re: We thank the reviewer for capturing this error. We have updated the formula in the Supplementary Note as “ $Var(\hat{\theta}) = X'Var(\hat{\beta})X + tr(Var(\hat{\beta})Var(\hat{X}))$ ” where the true values of X are unknown and therefore we replace them by their estimates \hat{X} .”

Reviewer #1 (Remarks to the Author):

Thank you to the authors for their thorough responses. I have no further comments.